# Analytical approach of synchronous and asynchronous update schemes applied to solving biological Boolean networks

Antonio Bensussen[1], J. Arturo Arciniega-González[2], Elena R. Álvarez-Buylla[3], Juan Carlos Martínez-García[1]*

**1** Departamento de Control Automático, Cinvestav-IPN, Ciudad de México, Mexico, **2** Programa Doctoral en Ciencias Biomédicas, Universidad Nacional Autónoma de México, Ciudad de México, Mexico, **3** Instituto de Ecología, Universidad Nacional Autónoma de México, Ciudad de México, Mexico,

* juancarlos.martinez@cinvestav.mx

## Abstract

Characterizing the minimum, necessary and sufficient components to generate the dynamics of a biological system has always been a priority to understand its functioning. In this sense, the canonical form of biological systems modeled by Boolean networks accurately defines the components in charge of controlling the dynamics of such systems. However, the calculation of the canonical form might be complicated in mathematical terms. In addition, computing the canonical form does not consider the dynamical properties found when using the synchronous and asynchronous update schemes to solve Boolean networks. Here, we analyze both update schemes and their connection with the canonical form of Boolean networks. We found that the synchronous scheme can be expressed by the Chapman-Kolmogorov equation, being a particular case of Markov chains. We also discovered that the canonical form of any Boolean network can be easily obtained by solving this matrix equation. Finally, we found that, the update order of the asynchronous scheme generates a set of functions that, when composed together, produce characteristic properties of this scheme, such as the conservation of fixed-point attractors or the variability in the basins of attraction. We concluded that the canonical form of Boolean networks can only be obtained for systems that use the synchronous update scheme, which opens up new possibilities for study.

## Introduction

Nowadays, there is a growing interest in understanding in depth the functioning of biological systems [1]. In this sense, the use of mathematical and computational models to represent such systems is one of the most novel approaches of study [1,2]. Indeed, this approach allows to understand and even to predict the functioning of biological systems at quantitative and qualitative levels [3,4]. Although there is no

**Data availability statement:** All relevant data are within the manuscript and its Supporting Information files.

**Funding:** Antonio Bensussen acknowledges the support from SECIHTI CBF-2025-G-494. Elena R. Álvarez-Buylla acknowledges the support from UNAM-DGAPA PAPIIT IN211721 "Patrones genéricos y sistémicos de la diferenciación y la proliferación en los nichos de células troncales: Raíz de Arabidopsis thaliana como sistema de estudio teórico-experimental" and from SECIHTI CBF-2025-G-344. Juan Carlos Martínez-García acknowledges the support from SECIHTI CF-2019/194186 "Biología matemática y computacional de sistemas médicos: modulación preventiva de la emergencia y progresión de enfermedades crónico-degenerativas". The funders had no role in study design, data collection and analysis, decision to publish, or preparation of the manuscript.

**Competing interests:** The authors have declared that no competing interests exist.

limitation that restricts the type of models that should be used to represent a particular biological system, it is common to choose the modeling framework based on the information available in the specialized literature or in direct experimental data [5]. In contexts where only the molecular interactions are known, and it is not possible to estimate any quantitative parameters, the recommended modeling framework is Boolean networks, since they directly transform the experimental interactions into logical functions [5].

Boolean networks are a flexible tools that allow the study of all types of biological systems, such as plant development [6], immunological response [7,8], neoplasia development [9], pathogen-host interactions [10], among others. To work with these models, it is important to mention that, each of the molecular species involved in a biological process becomes a node of the network. These nodes can only have two states: "*on*" and "*off*", which corresponds to the set $\mathbb{B} := \{0, 1\}$, where 0 is "*off*" and 1 is "*on*". All experimentally reported interactions that can modify the state of each node is used to create a logical function to determine the future state of the node studied. That is: $x_i(t+1) = f_i(x_1(t), x_2(t), \ldots, x_n(t))$, where $x_i(t+1)$ is the future state of node $x_i$, and $f_i$ represents all logical interactions of $x_1(t), x_2(t), \ldots, x_n(t)$ to determine the state of node $x_i$ [11].

As a result of the temporal evolution of this type of dynamical systems, configurations of time-invariant systems belonging to the state space of the network ($\Omega := \mathbb{B}^n$) appear, which are known as attractors [11], which can be composed of a single state that remains fixed when $t \to \infty$, or by several states that are repeated cyclically. The first type of attractors is known as "*fixed-point attractors*", while the second type is known as "*cyclic attractors*" [11]. The set of attractors obtained for Boolean networks is known as "*landscape of attractors*", and it can be found using either analytical or numerical approximations. Concerning to the numerical approaches, a well-known method to compute the solution of Boolean networks is the "*brute force*", which uses all configurations of the network state space ($\Omega$) to update the network, in order to seek which attractor is reached. To update the network, all logic rules can be evaluated at the same time, i.e., synchronously, or the evaluation can be done by assigning a specific update time to each node, that is, asynchronously [12]. Using both update strategies to solving networks with more than 30 nodes can be almost impossible, since the state space has $2^n$ possible configurations and all of them must be explored [13].

For this reason, analytical approaches have been developed to fully characterizing the dynamics of Boolean networks, regardless of the update scheme. In this sense, Liu and Cheng proposed a formalism based on the semi-tensor product of matrices, in which, through a series of algebraic transformations, they converted a set of logic rules of a Boolean network into a system of linear equations (14). The idea of this analytical approach focuses on defining the "*canonical form*" of Boolean networks, which reveals the minimum, necessary and sufficient structural components to obtain the dynamics of any Boolean system [14]. In practice, this approach is not easy to implement, since for small systems the help of software is required for its resolution [15]. In this sense, Wang *et al*. proposed the use of "*truth matrices*", to transform

Boolean networks into integers and then, to recover cyclic attractors easily [15]. The strategy of Wang *et al.* opens the possibility of changing the perspective of Boolean networks by using integers to solve them. In agreement with this idea, we showed that the landscape of attractors of any Boolean network can be visualized using integers in the Cartesian plane [13], and by studying this way of visualizing attractors, we found dynamical properties not previously described.

Thus, in this work we delved into the mathematical properties of this alternative visualization of the landscape of attractors of Boolean networks. Notably, we found that representing the landscape of attractors in the Cartesian plane is the key to easily obtain the canonical form of Boolean networks. Interestingly, our results showed that, the synchronous update scheme is a particular case of Markov chains. Furthermore, we were able to analytically study the asynchronous update scheme and, we found the origin of many of its distinctive dynamical features. Finally, we tested our results to solve real networks of biological systems, such as T cell differentiation, macrophage polarization, hematopoietic stem cell differentiation and the epithelial-to-mesenchymal transition in hepatocytes. We concluded that our approach is effective in analyzing Boolean networks grounded in experimental data.

## Materials and methods

### Computational implementation

The calculation of the attractors and its basins of attraction of all the networks presented in this work was performed using BoolNet [16] and GINSIM [17]. The implementation of the theorems developed in this work was done with the object-oriented language C# using Visual Studio 2022, community version.

## Results

### Canonical form of networks that only present fixed-point attractors

Studying the landscape of attractors of different types of Boolean networks allowed us to identify special properties of those systems whose solution only consists of fixed-point attractors [13]. In this sense, for such systems, their landscape of attractors behaves as a single Boolean function, as we previously demonstrated [13]. Now, we recapitulate this fact in the following lemma.

**Lemma 1.** Let $F : \mathbb{B}^n \to \mathbb{B}^n$ be a Boolean network of $n$ nodes updated synchronously, with $\Omega$ as its state-space. If $F$ only have fixed-point attractors as solution and if $F$ is converted into integer numbers, then their attractor landscape is equivalent to a function.

**Proof**. If the solution of $F$ only consists of fixed-point attractors, it means that $\exists P \subseteq \Omega$ such that, $\forall p \in P,\ F(t, p) = p$ for every $t > 0$. Since $F$ is updated synchronously, then $\forall u \in \Omega,\ \exists p \in P$, such that $F(t, u) = p$ when $t \to \infty$. Redefining the state-space as $\Omega := \left\{ w \in \mathbb{Z} : 0 \leq w < 2^n \right\}$, it is clear that, starting from any element in the domain, i.e., $u \in \Omega \subseteq \mathbb{Z}$, when $t \to \infty$, it will be reached a fixed-point attractor in the codomain $p \in P \subseteq \Omega$. Consequently, this defines a function $H : \Omega \to \Omega$ in the Cartesian plane, whose image will be $P$ ∎

In other words, this Lemma states that if a Boolean network only has fixed-point attractors as a solution, then, its landscape of attractors can be defined as a function $H$ that has the state space as domain, and its image corresponds to the set of fixed-point attractors. Now, if we convert the elements of the state space to integers, we can represent the solution of such a Boolean network on a Cartesian plane. Let's illustrate this statement with the following example:

**Example 1**. Consider the following Boolean network:

$$\begin{cases} x_1^+ = x_2 x_3 \\ x_2^+ = x_3 \\ x_3^+ = x_3 \end{cases},$$

(E. 1)

Solving the network (E. 1) by brute force, we obtain the results shown in Fig 1A. Note that, assuming that the initial configurations and attractors are binary numbers, we can convert them to integers (Fig 1A). In this way, it is easy to plot the solution of (E. 1) on a Cartesian plane as Fig B shows. Observe that the attractor landscape of (E.1) behaves as a logical function for which, from the data obtained by the brute force method (Fig 1A), we can calculate its disjunctive normal form by adding the min-terms that produce 1 for the three nodes [18], that is:

$$\begin{cases} x_1^+ = \bar{x}_1\bar{x}_2x_3 + \bar{x}_1x_2x_3 + x_1\bar{x}_2x_3 + x_1x_2x_3 \\ x_2^+ = \bar{x}_1\bar{x}_2x_3 + \bar{x}_1x_2x_3 + x_1\bar{x}_2x_3 + x_1x_2x_3 \\ x_3^+ = \bar{x}_1\bar{x}_2x_3 + \bar{x}_1x_2x_3 + x_1\bar{x}_2x_3 + x_1x_2x_3 \end{cases}$$

Simplifying these expressions by using Boolean algebra, we obtain:

$$\begin{cases} x_1^+ = x_3 \\ x_2^+ = x_3 \\ x_3^+ = x_3 \end{cases},$$

(E. 2)

Note that the solution of (E. 2) by brute force is identical to the original network (E. 1) (Fig. 1C). This implies that, the network (E. 2) is a simplified version of the network (E.1) and share the same landscape of attractors.

Thus, the main implication of Lemma 1 is that the canonical form of a Boolean network that only has fixed-point attractors can be observed in the Cartesian plane, and its algebraic expression can be found by using classical procedures for computing the canonical form of any logical function, such as the disjunctive normal form.

## Dynamical properties of cyclic attractors

The above procedure for finding the canonical form of a Boolean network can only be applied to networks whose solution consists exclusively of fixed-point attractors. This is because cyclic attractors have special characteristics, typical

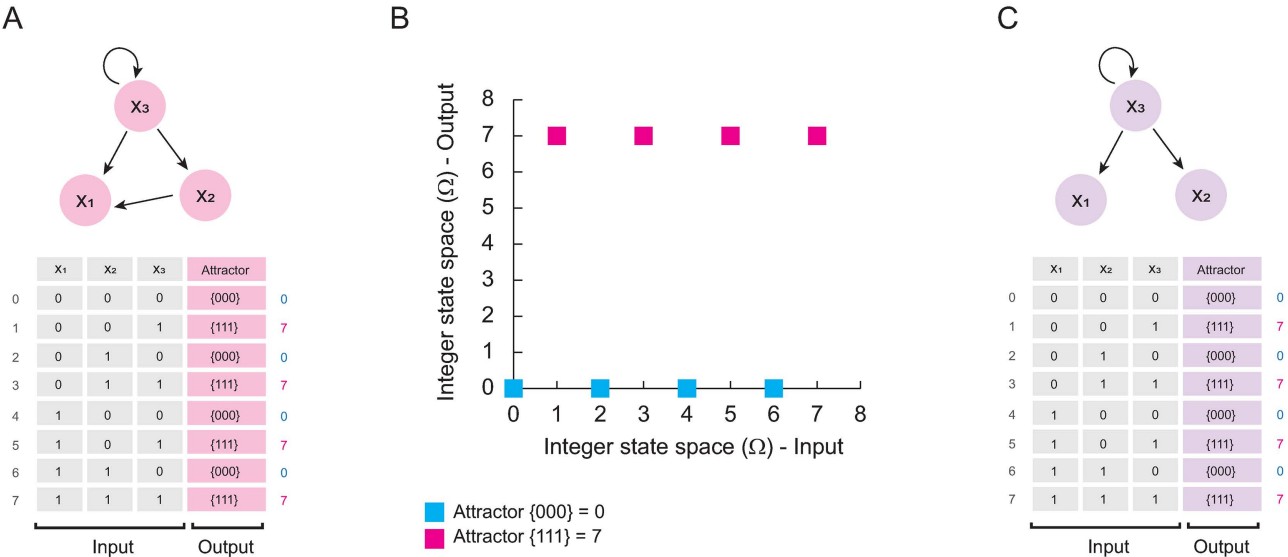

**Fig 1. Simplification of networks that only have fixed-point attractors.** In panel (A) is shown the network (E. 1) and its synchronous solution calculated with BoolNet [16] and GINSIM [17]. Note that the input and output settings are equivalent to integers. In panel (B) is shown the projection of the landscape of attractors calculated in the last panel. On the other hand, in panel (C) is shown the network (E. 2) with its respective computational solution. Note that both networks have the same landscape of attractors, which implies that both are dynamically equivalent.

of Boolean relations rather than functions, as we previously demonstrated [13]. Now, we recapitulated this result in the following proposition.

**Proposition 2.** Let $F : \mathbb{B}^n \to \mathbb{B}^n$ be a Boolean network of $n$ nodes updated synchronously, with $\Omega$ as it state-space. If $C$ is a cyclic attractor of $F$, and if it is converted to integers, then $C$ can be plotted as a relation in the Cartesian plane.

**Proof**. If $C$ is a cyclic attractor of $F$, it implies that $\forall c \in C \subseteq \Omega$, $F(t + \tau, c) = c$, for $t > 0$, and $\tau \in \mathbb{Z}$. Since each attractor has a basin of attraction $B = \{b_1, b_2, \ldots, b_m\} \subseteq \Omega$, when $t \to \infty$ every element of $B$ reaches each element of $C$, which means that the elements of the basin of attraction and the orbit of the cyclic attractor are related as follows:

$$R = \{(b_1, c_1), (b_1, c_2), \ldots, (b_1, c_s), (b_2, c_1), (b_2, c_2), \ldots, (b_2, c_s), \ldots, (b_m, c_1), (b_m, c_2), \ldots (b_m, c_s)\}$$

Redefining the state-space as $\Omega := \{w \in \mathbb{Z} : 0 \leq w < 2^n\}$, occurs that for each element in the domain $u \in \Omega \subseteq \mathbb{Z}$, when $t \to \infty$, exists more than one element in the codomain, given by all elements of $C \subseteq \Omega$, which defines a relation $R \subseteq \Omega^2$ in the Cartesian plane■

Conceptualizing cyclic attractors as Boolean relations allows us to study this type of attractors with alternative algebraic approaches. In this sense, Bañares and colleagues proposed that, it is possible to study a Boolean relation by decomposing it into compatible functions [19]. The idea is to distribute all the interactions of a relation into a set of functions, as shown in Fig 2A [19]. Taking this proposal as a direct antecedent, we stated the following theorem.

**Theorem 3.** Let $R$ be a Boolean relation that describes the orbit of a cyclic attractor $C$ and let $c_1, c_2, c_3 \in C$. If element $c_1$ is related to element $c_2$, and element $c_2$ is related to element $c_3$, and element $c_3$ is related to element $c_1$ (i.e.,

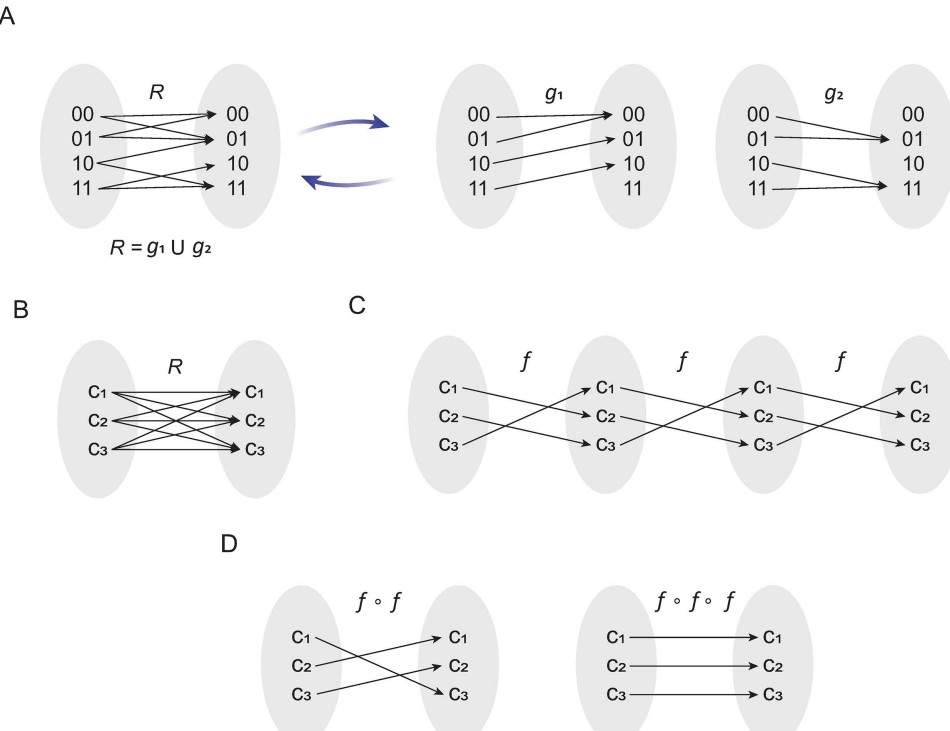

**Fig 2. Dynamical properties of cyclic attractors.** In panel (A) is shown the idea proposed by Bañares *et al* [21], in which a relation R is divided into compatible functions $g_1$ and $g_2$. In panel (B) is shown the typical interactions between elements of the orbit of a cyclic attractor **C**. In panel (C) is shown the definition of the compatible function **f**, and its own composition in order to define $(\mathbf{f} \circ \mathbf{f})$ and $(\mathbf{f} \circ \mathbf{f} \circ \mathbf{f})$, presented in panel **(D)**. Note that a cyclic attractor can be obtained from a single function **f**.

$c_1 R c_2 \wedge c_2 R c_3 \wedge c_3 R c_1$) then, exists a compatible function $f$ such that, the orbit represented by $R$ can be obtained by the union of compositions of the compatible function $f$. In other words:

$$R = f \cup (f \circ f) \cup (f \circ f \circ f)$$

**Proof.** Suppose that

$R := \left\{ (c_1, c_1), (c_1, c_2), (c_1, c_3), (c_2, c_1), (c_2, c_2), (c_2, c_3), (c_3, c_1), (c_3, c_2), (c_3, c_3) \right\}$,

note that $R$ fulfils $c_1 R c_2 \wedge c_2 R c_3 \wedge c_3 R c_1$ since $(c_1, c_2), (c_2, c_3), (c_3, c_1) \in R$ (Fig 2B). Then, we might propose $f := \left\{ (c_1, c_2), (c_2, c_3), (c_3, c_1) \right\}$ (Fig 2C), which is a compatible function of $R$. Observe that $(f \circ f) = \left\{ (c_1, c_3), (c_2, c_1), (c_3, c_2) \right\}$ and $(f \circ f \circ f) = \left\{ (c_1, c_1), (c_2, c_2), (c_3, c_3) \right\}$ (Fig 2D) are also compatible functions of $R$, and therefore $R = f \cup (f \circ f) \cup (f \circ f \circ f)$ ∎

The most important implication of this theorem is that a cyclic attractor can be generated by the composition of a single function, which means that a cyclic attractor also has a canonical form.

## Matrix-based method to solve synchronous updated Boolean networks

An important hint to find the compatible function that originates a cyclic attractor is that, by composing such a function several times, we might reconstruct the cyclic attractor (Fig 2C). Hence, in this section we will analyze the connection between function composition and the synchronous update scheme. According to its classical definition, the synchronous update scheme $S$ starts from any initial condition in the state space $u_0 \in \Omega$, to determine the state of the network $F : \mathbb{B}^n \to \mathbb{B}^n$ at the next time step [11]. To illustrate this concept, observe that, the zero step of the synchronous update $S$ is equivalent to the initial starting condition; that is, $S(u_0, t = 0) = u_0$. On the other hand, to compute the first update of $S$, the Boolean network function $F$ must be evaluated with the initial condition $u_0$; that is: $S(u_0, t = 1) = F(u_0)$. For the next update step, it is necessary to take the result obtained at $t = 1$ to evaluate $F$; in other words, $S(u_0, t = 0) = F(F(u_0))$. Generalizing this process, we can say that:

$$S(u_0, t = 0) = u_0$$
$$S(u_0, t = 1) = F(u_0)$$
$$S(u_0, t = 2) = F(F(u_0))$$
$$S(u_0, t = 3) = F(F(F(u_0)))$$
$$\vdots$$

Observe that applying the same function twice is the same as composing the same function, i.e., $g(g(x)) = (g \circ g)(x)$, then we can rewrite the generalization of the synchronous update scheme as follows:

$$S(u_0, t = 0) = u_0$$
$$S(u_0, t = 1) = F(u_0)$$
$$S(u_0, t = 2) = (F \circ F)(u_0)$$
$$S(u_0, t = 3) = (F \circ F \circ F)(u_0)$$
$$\vdots$$

Therefore, the synchronous update scheme is originated by the iterative composition of the original network $F$. It is important to note that, the composition of two functions is equivalent to multiply the matrices associated with each function, i.e., $(h \circ g) = M_h \times M_g$. Considering that, in this case, the same function is composed as the time steps advance, we might redefine the synchronous update scheme $S$ as:

$$S(t, u_0) := (M_F)^t u_0, \quad \forall t \geq 1 \tag{E. 3}$$

Where $M_F$ is the matrix associated with $F$. Remarkably, this alternative definition of the synchronous update scheme is the Chapman-Kolmogorov equation [20]].

Now, in order to find $M_F$, the Boolean network must be converted into a function that has an associated matrix. In this direction, Wang and colleagues propose to use the logical rules of a Boolean network to create a function that relates each of the elements of the network state space to a corresponding image, to work on it as if it were a function defined in integers [15]. We expanded this idea by using the function defined in integers to calculate $M_F$. We will illustrate this procedure with the following example.

**Example 2**: Let $F : \mathbb{B}^2 \to \mathbb{B}^2$ be a Boolean network given by:

$$\begin{cases} x_1^+ = x_1 + x_2 \\ x_2^+ = \bar{x}_1\bar{x}_2 + x_1x_2 \end{cases}$$

(E. 4)

If we use all the states of $\Omega$ to define a function $f^* : \Omega \to \Omega$, we obtain the result shown in Fig 3A. Converting $\Omega$ to integers gives us the function $f^{**}$ shown in Fig 3B. Thus, we might define the matrix $M_F$ as follows:

$$M_F = \begin{bmatrix} 0 & 1 & 0 & 0 \\ 0 & 0 & 1 & 0 \\ 0 & 0 & 1 & 0 \\ 0 & 0 & 0 & 1 \end{bmatrix}$$

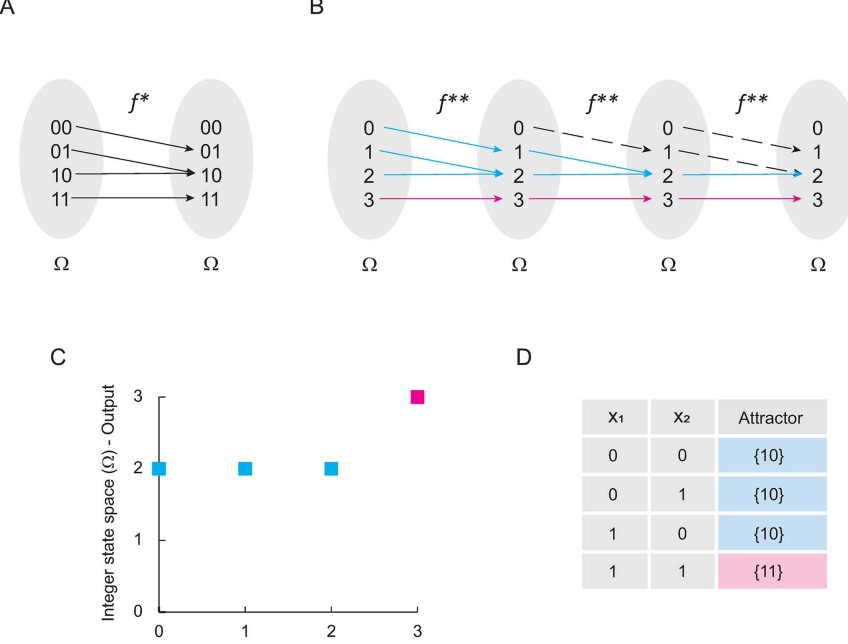

**Fig 3. Solving Boolean networks with only fixed-point attractors.** In panel (A) is shown the Boolean function **f**\* originated by the network (E. 4). In panel (B) is shown **f**\* transformed into integers. Blue lines indicate all trajectories from the domain of **f**\*\* to the attractor found in 2. Pink lines indicate the trajectory from the domain of **f**\*\* to attractor 3. Black dashed lines indicate other interactions excluded from trajectories originated by the iterative composition of **f**\*\*. In panel (C) is shown de Cartesian plane that represents the solution of **f**\*\*, while in panel (D) is shown the computational solution of the original network. Note that panels (C) and (D) represent the landscape of attractors of (E. 4).

Where the columns indicate the outputs and the rows the inputs of the function $f^{**}$. In consequence, the matrix expression equivalent to the synchronous method for the network $F$ is given by:

$$S(t, u_0) = \begin{bmatrix} 0 & 1 & 0 & 0 \\ 0 & 0 & 1 & 0 \\ 0 & 0 & 1 & 0 \\ 0 & 0 & 0 & 1 \end{bmatrix}^t u_0,$$

(E. 5)

Note that (E. 5) is the temporal transition matrix of jump described by the Chapman-Kolmogorov equation [22]. This interesting observation allowed us to obtain the stationary solution of this system ($S^*$) as a classical Markov chain, i.e.:

$$S^* = \lim_{t \to \infty} S(t),$$

(E. 6)

Then,

$$S^* = \begin{bmatrix} 0 & 0 & 1 & 0 \\ 0 & 0 & 1 & 0 \\ 0 & 0 & 1 & 0 \\ 0 & 0 & 0 & 1 \end{bmatrix}$$

Interestingly, this matrix corresponds to the landscape of attractors represented by the Cartesian plane shown in Fig 3C, and it is the solution of the network (E. 5) as shown in Fig 3D. Therefore, the synchronous update scheme for Boolean networks is a particular case of Markov chains.

**Special properties of the landscape of attractors**

The unexpected connection of the landscape of attractors with Markov chains allowed us to extrapolate interesting properties, such as, the existence of a stationary solution and the existence of periodic solutions [21]. The Example 2, showed that, for fixed-point attractors, the image of the landscape of attractors ($\Omega_Z^2$) is simply the stationary solution of $S(t)$, in other words:

$$\Omega_Z^2 = \lim_{t \to \infty} S(t),$$

(E. 7)

Concerning to periodic solutions of $S(t)$, we noted that $\Omega_Z^2$ can be obtained by adding the matrix calculated for each time-step contained in the period of a cyclic attractor. Let's see this point in the following example.

 **Example 3**: Let $F : \mathbb{B}^2 \to \mathbb{B}^2$ a Boolean network given by:

$$\begin{cases} x_1^+ = x_2 \\ x_2^+ = x_1 \end{cases},$$

(E. 8)

Using the logic rules of (E. 8) to generate a Boolean function $f^*$ (Fig 4A), and then transforming it into integers (Fig 4B) we obtain $f^{**} = \{(0,0),\ (1,2), (2,1), (3,3)\}$. Note that this function fulfils Theorem 3, since $(1,2), (2,1) \in f^{**}$, which implies that it generates a cyclic attractor $1 \to 2 \to 1$. Calculating the matrix expression of the synchronous method for the network $F$, we have:

$$S(t) = \begin{bmatrix} 1 & 0 & 0 & 0 \\ 0 & 0 & 1 & 0 \\ 0 & 1 & 0 & 0 \\ 0 & 0 & 0 & 1 \end{bmatrix}^t,$$

(E. 9)

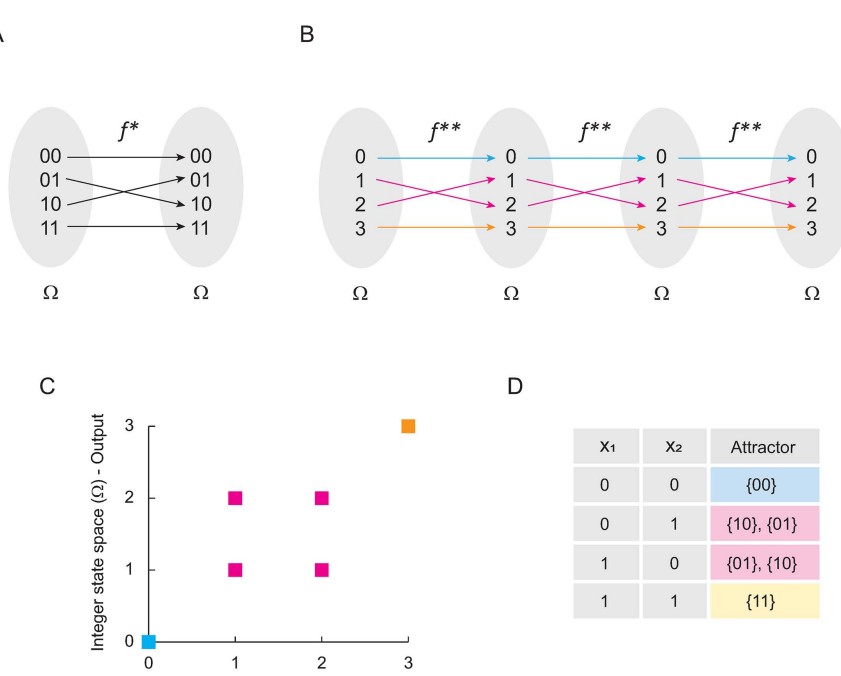

**Fig 4. Solving Boolean networks with periodic behavior.** In panel (A) is shown the function $\mathbf{f^*}$ originated by the Boolean network (E. 8). In panel (B) is shown $\mathbf{f^*}$ transformed into integers. Blue arrows indicate all trajectories from the domain of $\mathbf{f^{**}}$ to the attractor found in 0. Pink arrows indicate the trajectory from the domain of $\mathbf{f^{**}}$ to the cyclic attractor {1, 2}. Yellow arrows indicate the trajectory from the domain of $\mathbf{f^{**}}$ to the fixed-point attractor 3. In panel (C) is shown de Cartesian plane that represents the landscape of attractor of $\mathbf{f^{**}}$, and in panel (D) is shown the computational solution of the original network.

Evaluating $S(t)$ at $t = 2,\ 3,\ 4$, we obtain:

$$S(2) = \begin{bmatrix} 1 & 0 & 0 & 0 \\ 0 & 1 & 0 & 0 \\ 0 & 0 & 1 & 0 \\ 0 & 0 & 0 & 1 \end{bmatrix}$$

$$S(3) = \begin{bmatrix} 1 & 0 & 0 & 0 \\ 0 & 0 & 1 & 0 \\ 0 & 1 & 0 & 0 \\ 0 & 0 & 0 & 1 \end{bmatrix}$$

$$S(4) = \begin{bmatrix} 1 & 0 & 0 & 0 \\ 0 & 1 & 0 & 0 \\ 0 & 0 & 1 & 0 \\ 0 & 0 & 0 & 1 \end{bmatrix}$$

Note that $S(2) = S(4)$, and consequently, to calculate $\Omega_Z^2$ (Fig 4C) we need to add all periodic images that correspond to the cyclic attractor, such as:

$$S(2) + S(3) = \begin{bmatrix} 1 & 0 & 0 & 0 \\ 0 & 1 & 0 & 0 \\ 0 & 0 & 1 & 0 \\ 0 & 0 & 0 & 1 \end{bmatrix} + \begin{bmatrix} 1 & 0 & 0 & 0 \\ 0 & 0 & 1 & 0 \\ 0 & 1 & 0 & 0 \\ 0 & 0 & 0 & 1 \end{bmatrix} = \begin{bmatrix} 1 & 0 & 0 & 0 \\ 0 & 1 & 1 & 0 \\ 0 & 1 & 1 & 0 \\ 0 & 0 & 0 & 1 \end{bmatrix}$$

Observe that, such a matrix corresponds exactly to the computer-calculated landscape of attractors presented in Fig 4D. Therefore, the logical addition of the matrices for the time steps belonging to the period ($T$) of a cyclic attractor is equal to the image of the landscape of attractors seen in the Cartesian plane ($\Omega_Z^2$), i.e.,:

$$\Omega_Z^2 = \sum_{t \in T} S(t), \tag{E. 10}$$

### Fixed-point attractors and its basins of attraction are easy to characterize

In Example 3 it can be observed that the function $f_{**}$ allows to identify at a glance the fixed-point attractors, since they only interact with themselves and in consequence, they are reflexive elements, i.e., elements related with themselves. Formally, we shall say:

**Theorem 4**: Let $H : \mathbb{B}^n \to \mathbb{B}^n$ be a synchronously updated Boolean network of $n$ nodes and $\Omega$ as it states space. If $H$ is converted into $H^* : \Omega \to \Omega$ and it has reflexive elements, then these elements are fixed-point attractors of $H$.

**Proof**: If $H^*$ has reflexive elements, it implies that $\exists x, y \in \Omega$, such that $(x, x), (y, y) \in H^*$. Considering that, the synchronous update scheme can be calculated by making composition of $H^*$ as a function of the time-steps, we might

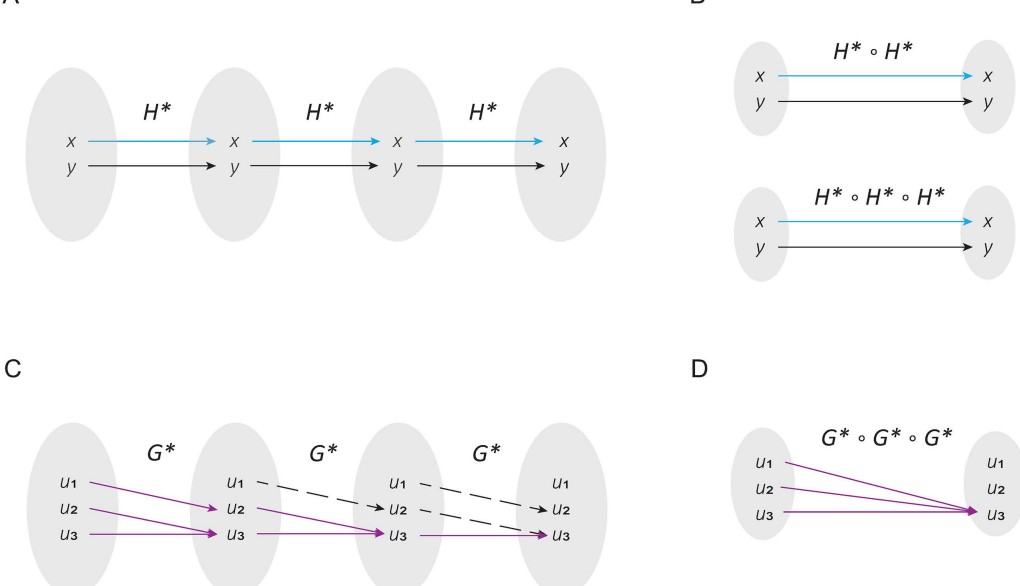

**Fig 5. Finding fixed-point attractors and its basins of attraction.** In panel (A) is defined $\mathbf{H}^*$ and the trajectories originated from the iterative composition of this function. Blue lines represent the trajectories to $\mathbf{x}$ and black lines represent all trajectories to $\mathbf{y}$. In panel (B) is presented the formal definition of $(\mathbf{H}^* \circ \mathbf{H}^*)$ and $(\mathbf{H}^* \circ \mathbf{H}^* \circ \mathbf{H}^*)$. Note that both composed functions are the same as the original, which proves that $\mathbf{x}$ and $\mathbf{y}$ remain fixed despite the iterative composition of $\mathbf{H}^*$. In panel (C) is defined $\mathbf{G}^*$, where purple lines represent all trajectories from the domain of $\mathbf{G}^*$ to the attractor $\mathbf{u}_3$. Black dashed lines represent all interactions that are excluded from these trajectories. In panel (D) is presented $(\mathbf{G}^* \circ \mathbf{G}^* \circ \mathbf{G}^*)$, and note that $\mathbf{u}_1$ and $\mathbf{u}_2$ eventually converge to $\mathbf{u}_3$, which illustrates the concept of basin of attraction.

see that $(x, x), (y, y) \in (H^* \circ H^*)$, and $(x, x), (y, y) \in (H^* \circ H^* \circ H^*)$ and so on for any time-step (Figs 5A and 5B). This implies that $x$ and $y$ always arrives to their own values for any time-step, which is the definition of a fixed-point attractor ∎

Concerning to the basins of attraction, these can be identified using the approach described above to obtain the orbit of a cyclic attractor. This principle is stated in the following proposition.

**Proposition 5**: Let $G^*$ a synchronously updated Boolean network of $n$ nodes redefined as a function of integers, and let $u_1, u_2, u_3 \in Dom(G^*)$. If the element $u_3$ is an attractor and $(G^* \circ G^* \circ G^*)(u_{1,2}) = u_3$, then $u_1$ and $u_2$ belongs to the basin of attraction of $u_3$.

**Proof**: If $u_3$ is an attractor, it implies that $(u_3, u_3) \in G^*$. Suppose that we defined the function $G^* := \{(u_1, u_2), (u_2, u_3), (u_3, u_3)\}$(Fig 5C). Note that $(G^* \circ G^* \circ G^*) = \{(u_1, u_3), (u_2, u_3), (u_3, u_3)\}$ (Fig 5D), which implies that, while $t \to \infty$, $u_1$ and $u_2$ converge to the attractor $u_3$. Thus, $u_1$ and $u_2$ belongs to the basin of attraction of $u_3$ by definition ∎

An important implication of this theory is that the basin of attraction, and the attractors in general, can be identified by a systematic search in a function defined on integers. Remarkably, *this idea also applies to cyclic attractors.*

## Canonical form of synchronous updated networks

Now, we have all elements required to state a method for computing the canonical form of any synchronously updated Boolean network. Our proposal is based on the fact that, by the definition presented in this paper, the synchronous update scheme $S(t)$ uses a single function to generate the entire dynamics of the network. Thus, if we evaluate $S(t)$ when $t$ is large, i.e., $S^* = \lim_{t \to \infty} S(t)$. Then, we might use this function to compute the canonical form with any of the valid approaches for logical functions. Importantly, if the network converted into a function $F^{**}$ initially has a cycle, the image of $S(t)$ as $t \to \infty$ must still contain the same cycle to calculate its canonical form. Let's illustrate this procedure with the following example.

**Example 4**: Let $F : \mathbb{B}^2 \to \mathbb{B}^2$ be a Boolean network given by:

$$\begin{cases} x_1^+ = x_1 + x_2 \\ x_2^+ = \overline{x}_2 \end{cases},$$

(E. 11)

After using the network (E. 11) to create a Boolean function $f^*$ (Fig 6A), and then transforming it into integers (Fig 6B), the expression $S(t)$, we obtain:

$$S(t) = \begin{bmatrix} 0 & 1 & 0 & 0 \\ 0 & 0 & 1 & 0 \\ 0 & 0 & 0 & 1 \\ 0 & 0 & 1 & 0 \end{bmatrix}^t$$

Note that $f^{**}$ does not have any reflexive element, and by Theorem 4, this network does not have fixed-point attractors. On the other hand, by Theorem 3, the network $F$ contains a cycle given by $2 \to 3 \to 2$, since $(2, 3), (3, 2) \in f^{**}$ (Fig 6B). Now, to approximate $S(t)$ when $t \to \infty$, we could evaluate $t = 51$, then:

$$S(51) = \begin{bmatrix} 0 & 0 & 0 & 1 \\ 0 & 0 & 1 & 0 \\ 0 & 0 & 0 & 1 \\ 0 & 0 & 1 & 0 \end{bmatrix}$$

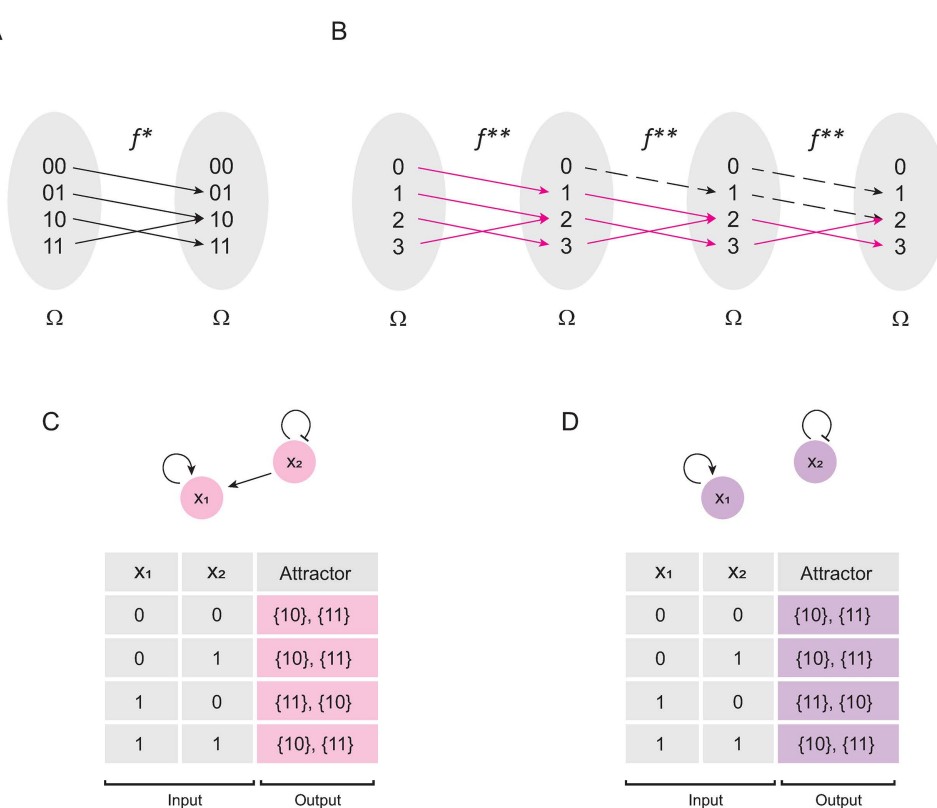

**Fig 6. Simplification of networks that only have cyclic attractors.** In panel (A) is shown the result of using the network (E. 11) to generate a Boolean function **f***. In panel (B) is shown the conversion of **f*** to integers, where pink lines indicate all trajectories that converge to cyclic attractor {2, 3}. In panel (C) is presented the network (E. 11) together with its computational solution. In panel (D) is presented the network (E. 12) together with its computational solution. Note that both networks have an identical solution, which proves that (E. 12) is a simplified version of (E. 11).

Reconfiguring the Boolean function equivalent to $S(51)$, we obtain:

$$f = \{(0, 3), (1, 2), (2,3), (3,2)\}$$

Note that, the cycle $2 \to 3 \to 2$ is still present, since $(2,3), (3,2) \in f$. Then, calculating the disjunctive normal form for (E.11), we have:

$$\begin{cases} x_1^+ = \bar{x}_1\bar{x}_2 + \bar{x}_1 x_2 + x_1\bar{x}_2 + x_1 x_2 \\ x_2^+ = \bar{x}_1\bar{x}_2 + x_1\bar{x}_2 \end{cases}$$

Simplifying this canonical form, we obtain:

$$\begin{cases} x_1^+ = 1 \\ x_2^+ = \bar{x}_2 \end{cases} \tag{E. 12}$$

Which is the simplified version of the network (E. 11), as can be corroborated by comparing their computer-calculated image of the landscape of attractors (Figs 6C and 6D).

## Simplified method for calculating the canonical form of synchronous updated networks

The method described in the previous section can be complicated for large networks, since it requires $2^n \times 2^n$ matrices. Considering this point, we now propose an alternative method for large networks, in which we used the above theorems to solve synchronously updated networks and to find their canonical form. Regarding to solving Boolean networks synchronously, it must be first created an integer-defined function equivalent to a Boolean network. Then, by performing a search considering Theorems 3 and 4, the cyclic and fixed-point attractors can be detected. Finally, applying Proposition 5, one must connect to all the elements that converge to a cyclic or fixed-point attractor, that is, the elements of its basin of attraction. To find the canonical form of a Boolean network, once its solution is found, the trajectories within the basins of attraction must be compacted so that all its elements converge on the attractor, in order to apply any procedure for obtaining canonical form for Boolean functions. Let's illustrate the above procedure with the following example.

**Example 5**: Consider the following Boolean network:

$$\begin{cases} x_1^+ = x_1 \\ x_2^+ = x_2 + x_3 \\ x_3^+ = (x_1 + \bar{x}_3)(x_2 + \bar{x}_3) \end{cases}$$

(E. 13)

Which computational solution is shown in Fig 7A. Converting the network (E. 13) into a function $f^*$ (Fig 7B), we obtain:

$$f^{**} = \big\{(0,1),(1,2),(2,3),(3,2),(4,5),(5,6),(6,7),(7,7)\big\}$$

Observe that $f^{**}$ is graphically represented in Fig 7C. Applying Theorems 3 and 4 we found a cyclic attractor in $2 \to 3 \to 2$, and a fixed-point attractor in $7$, since $(2,3),(3,2) \in f^{**}$ and $(7,7) \in f^{**}$. Now, as $(0,1),(1,2),(2,3),(3,2) \in f^{**}$, this implies that the basin of attraction of $2 \to 3 \to 2$ is composed by $B_{2,3} = \{0, 1, 2, 3\}$. Similarly, note that the basin of attraction of $7$ is defined by $B_7 = \{4, 5, 6, 7\}$. To calculate the canonical form of (E.13), we just need to redefine the function by compacting the interactions that converge on the attractors as follows:

$$f_\infty^{**} = \big\{(0,3),(1,2),(2,3),(3,2),(4,7),(5,7),(6,7),(7,7)\big\}$$

This function is graphically represented in Fig 7D. Next, we rewrote this function in binary numbers (Fig 7E) to apply the formalism of the disjunctive normal form calculation. As a result of the above, we obtain:

$$\begin{cases} x_1^+ = x_1\bar{x}_2\bar{x}_2 + x_1\bar{x}_2x_3 + x_1x_2\bar{x}_3 + x_1x_2x_3 \\ x_2^+ = \bar{x}_1\bar{x}_2\bar{x}_3 + \bar{x}_1\bar{x}_2x_3 + \bar{x}_1x_2\bar{x}_3 + \bar{x}_1x_2x_3 + x_1\bar{x}_2\bar{x}_3 + x_1\bar{x}_2x_3 + x_1x_2\bar{x}_3 + x_1x_2x_3 \\ x_3^+ = \bar{x}_1\bar{x}_2\bar{x}_3 + \bar{x}_1x_2\bar{x}_3 + x_1\bar{x}_2\bar{x}_3 + x_1\bar{x}_2x_3 + x_1x_2\bar{x}_3 + x_1x_2x_3 \end{cases}$$

Simplifying this form, we obtain:

$$\begin{cases} x_1^+ = x_1 \\ x_2^+ = 1 \\ x_3^+ = x_1 + \bar{x}_3 \end{cases}$$

(E. 14)

Note that, the computational solution of both networks is identical (Figs 7A and 7F). Therefore, solving Boolean networks using the synchronous update scheme and finding their canonical form can be summarized in a classical search algorithm.

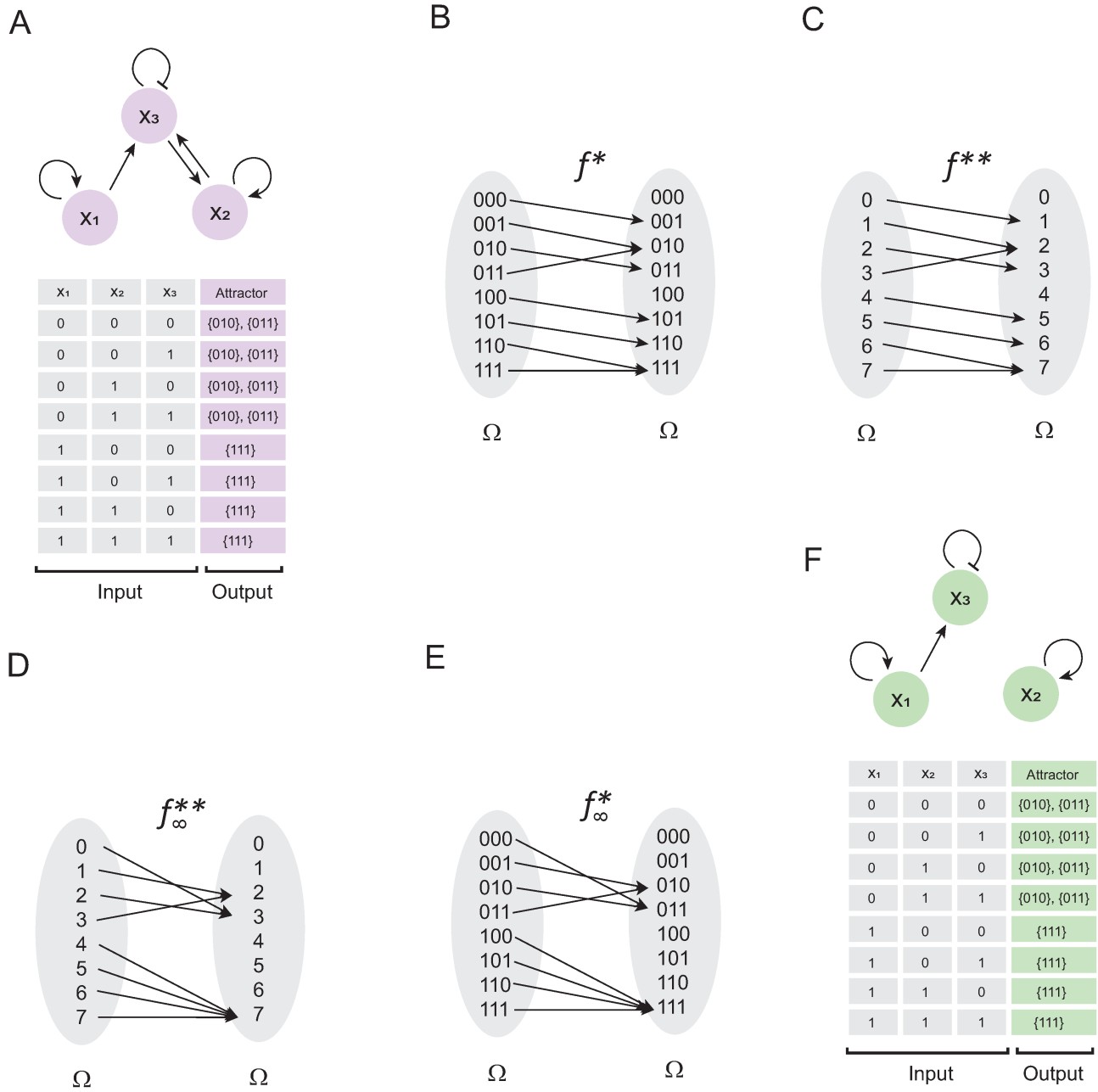

**Fig 7. Simplification of Boolean networks in general.** In panel (A) is presented the Boolean network (E. 13) with its computational solution calculated with BoolNet [16] and GINSIM [17]. In panel (B) is presented the Boolean function **f**∗, obtained by using the logic rules of (E. 13). In panel (C) is shown the transformation of **f**∗ to integers. In panel (D) is presented the function **f**∗∗ with all their interactions sent directly to the attractor to which they naturally converge, and in panel (E) is shown the conversion of this function to binary numbers. Finally, in panel (F) is shown the network generated from **f**∗∞, which corresponds to the network (E. 14). Note that (E. 14) is a simplified version of (E. 13).

## Matrix formalism of asynchronous update scheme

Since its conceptualization by René Thomas to study biological systems [22,23], the asynchronous update scheme has generated multiple studies to describe and characterize its mathematical properties. Some interesting properties of this

update scheme include the fact that negative feedback loops can generate homeostasis with or without periodicity, while positive feedback loops generate multiple alternative stationary states [22–24]. Later studies characterized new properties of the asynchronous update scheme, such as that some cyclic attractors could not be recovered using this update scheme, or that some elements of the basins of attraction changed depending on the update order of the nodes [25]. These studies also showed that fixed-point attractors are invariant to the update scheme used to analyze Boolean networks [25]. Motivated to understand the origin of these disjoint properties and to expand the analytical focus of studying the asynchronous update scheme, we decided to apply our matrix approach to study it.

Interestingly, we found that the asynchronous update scheme is capable of generating a set of functions that iterate based on the gene update order. Consequently, *it is not possible to obtain a unique canonical form for this update scheme, since its matrix definition changes depending on the gene update order*. Let's illustrate this point with the following example:

**Example 6**: Consider the Boolean network (E. 8) presented in the Example 3. There, we obtained $f^{**} = \{(0,0),\ (1,2), (2,1), (3,3)\}$ as the function associated with (E. 8) (Fig 8A), and its computational solution is presented in Fig 8B. Nevertheless, if the node $x_1$ is updated every two time-steps, and the node $x_2$ is updated every three time-steps, we will notice that several functions can be generated when we evaluate the system from $t = 0$ to $t = 6$ (Fig 8B). Individually, such functions are given by:

$$f_0 = \{(0,0),\ (1,1),(2,2),(3,3)\}$$

$$f_1 = \{(0,0),\ (1,3),(2,0),(3,3)\}$$

$$f_2 = \{(0,0),\ (1,0),(2,3),(3,3)\}$$

$$f_3 = \{(0,0),\ (1,2),(2,1),(3,3)\}$$

Note that, by $(f_2 \circ f_1 \circ f_0)$ for the first three time-steps, the system image converges to fixed-point attractors (Fig 8C), fact that explains the change observed on its computational solution presented in Fig 8D. Interestingly, if we change the update order (Fig 8E), we will notice that, functions $f_0$, $f_1$, $f_2$, and $f_3$ reappear but in a different order than that observed with the first asynchronous update scheme, producing notable differences in its landscape of attractors (Fig 8F).

Interestingly, by $t = 2$, the image of the system composition has already converged on fixed-point attractors in both cases (Figures 8C and 8E). Note that, for this example of asynchronous update, the cyclic attractor reported for this network has disappeared, and the size and elements of the basins of attraction have varied without altering the presence of the fixed-point attractors (Figures 8D and 8F). Finally, considering that the solution of both update schemes only consists in fixed-point attractors, we shall calculate the disjunctive normal form for the landscape of attractors recovered in Fig 8D:

$$\begin{cases} x_1^+ = x_1\bar{x}_2 + x_1 x_2 \\ x_2^+ = \bar{x}_1 x_2 + x_1 x_2 \end{cases}$$

(E. 15)

We also performed the same calculation for the landscape of attractors presented in Fig 8F:

$$\begin{cases} x_1^+ = x_1\bar{x}_2 + x_1 x_2 \\ x_2^+ = x_1\bar{x}_2 + x_1 x_2 \end{cases}$$

(E. 16)

 

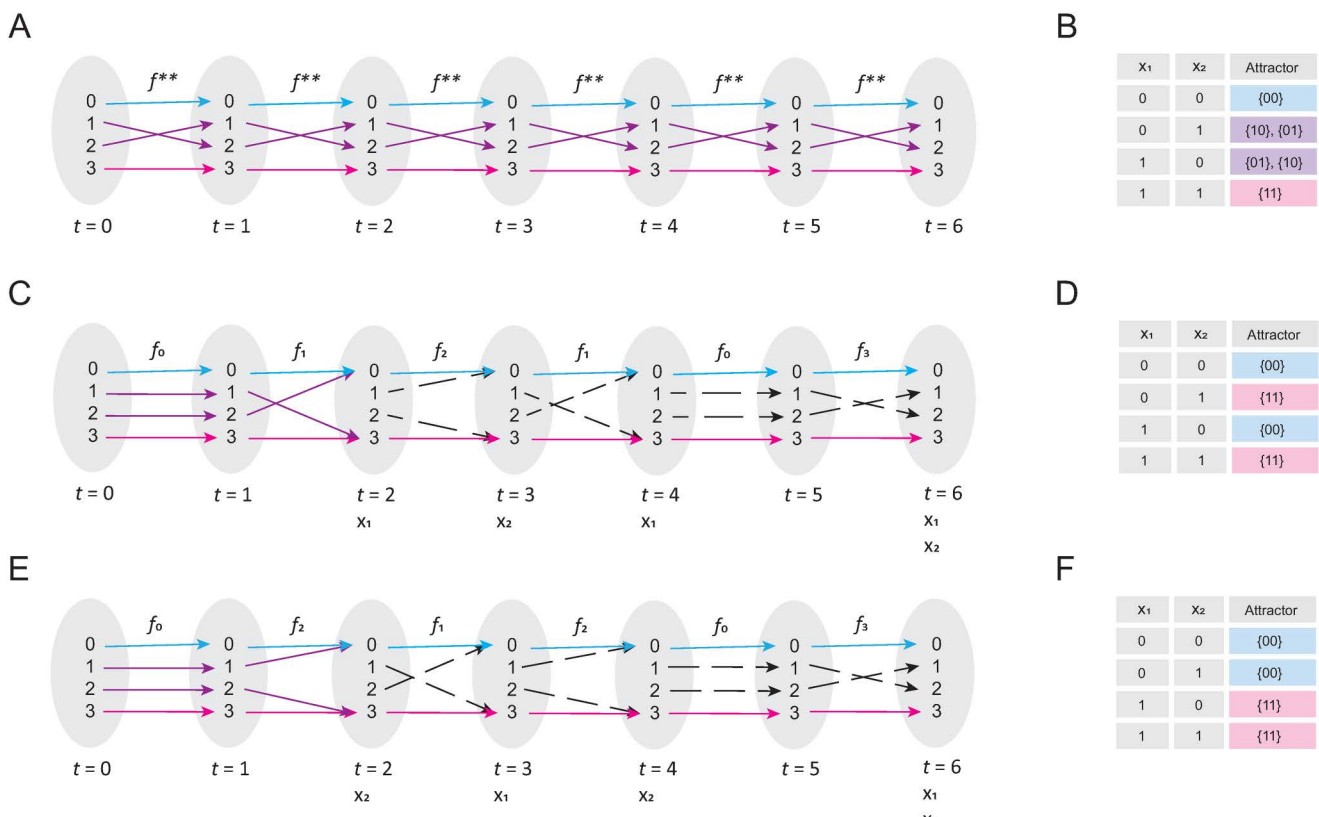

**Fig 8. Exploration of asynchronous update scheme.** Panel (A) shows the synchronous update sequence of the (E. 8) network, and panel (B) shows the computationally obtained attractor landscape of (E. 8). Blue and pink arrows indicate the trajectory from the domain of **f*** to the fixed-point attractors 0 and 3, respectively. Purple arrows indicate the trajectory from the domain of **f*** to the cyclic attractor {1, 2}. Panel (C) shows the update sequence of the (E. 8) network if the update time for $\mathbf{x}_1$ is two time-steps, and for $\mathbf{x}_2$ it is three-time steps. The dashed arrows show the interactions that are not part of the composition of functions that underlies the trajectory from the domain of **f***. Panel (D) shows the landscape of attractors calculated for the asynchronous updated described for panel (C). Panel (E) shows the result of inverting the asynchronous update order and panel (F) shows its landscape of attractors. Note that the update order generates functions that do not exist in the synchronous scheme, decouples the cyclic attractors, and alters the basins of attraction of the fixed-point attractors.

Simplifying (E. 15), we obtain:

$$\begin{cases} x_1^+ = x_1 \\ x_2^+ = x_2 \end{cases}$$

(E. 17)

Doing the same for (E. 16), we obtain:

$$\begin{cases} x_1^+ = x_1 \\ x_2^+ = x_1 \end{cases}$$

(E. 18)

Observe that none of these two simplified systems corresponds to the original form of the network (Fig 8B). Thus, it is not possible to obtain a unique canonical form for this update scheme, because asynchronous update scheme composes several functions using different orders, and the composition of functions is not commutative.

## Proof of concept on real biological networks

Now, we tested the effectiveness of our theorems to deal with real biological networks. To this end, we selected four previously published networks focused on studying different aspects of the functioning of the immune system. The first network was a 12-node network that models the differentiation of CD4 + T cells in response to microenvironments of different pro- and anti-inflammatory cytokines [8]. The second network was an 18-node network that represents the differentiation of CD8 + T cells in response to metabolic changes and cytokine combinations [7]. The third network was a 20-node network that models the differentiation of hematopoietic stem cells (HSCs) to give rise to the myeloid, lymphoid, and erythroid lineages [26]. The fourth selected network was a 15-node network that models the polarization of macrophages in response to cytokines present in the extracellular environment [27]. Finally, we selected a 23-node network that models the epithelial-to-mesenchymal transition in human hepatocytes proper of hepatocellular carcinoma [9].

These networks were implemented in BoolNet [16] and GINSIM [17] using the synchronous update scheme (Supporting information). On the other hand, to implement Theorem 4 to calculate the fixed-point attractors, a list of all the initial conditions ($Fx$) was first created for each of the networks, that is, lists with $2^n$ elements, where $n$ is the number of nodes in each network. Next, all the networks were evaluated with each of their initial conditions to create a new list of output elements ($Fy$). Finally, we searched in both lists elements that met the condition $Fx(i) = Fy(i)$, $i \in \mathbb{Z}$, since Theorem 4 mentions that this class of elements are by definition the fixed-point attractors of a Boolean network. This simple algorithm was implemented with the C# programming language in Visual Studio 2022 (Supporting information). As a result of these two procedures, we find that the implementation of Theorem 4 is able to find all the fixed-point attractors of the network, just like the BoolNet and GINSIM software (Fig 9), which demonstrates the effectiveness of the theory developed in this work.

## Solving large scale networks using the canonical form

Analytical approaches to working with large networks have been developed for years [28,29]. The central idea of these approaches is to divide a large network into connected modules, to find the semi-attractors of each module, and join them to obtain the global attractors [28,29]. However, this interesting methodology does not consider how to recover the basins of attraction. As we demonstrated in previous sections, the canonical form of a Boolean network contains the relevant information to fully recover the attractor landscape of the network, avoiding unnecessary interactions. For this reason, we decided to explore whether the canonical form of Boolean networks can improve the aforementioned methodology to solve large scale networks by recovering the basins of attraction. To explain our proposal, we will use the following nine-node Boolean network:

$$\begin{cases} x_1^+ = x_1 \\ x_2^+ = x_1 \\ x_3^+ = x_2 \\ x_4^+ = x_4 \\ x_5^+ = x_4 \\ x_6^+ = x_5 \\ x_7^+ = x_6 \\ x_8^+ = x_7 \\ x_9^+ = x_8 \end{cases}$$

(E. 19)

If we divide this 9-node network into 3-node modules, we can rewrite them as:

$$\begin{cases} x_1^+ = x_1 \\ x_2^+ = x_1 \\ x_3^+ = x_2 \end{cases} \qquad \begin{cases} x_4^+ = x_4 \\ x_5^+ = x_4 \\ x_6^+ = x_5 \end{cases} \qquad \begin{cases} x_7^+ = x_6 \\ x_8^+ = x_7 \\ x_9^+ = x_8 \end{cases}$$

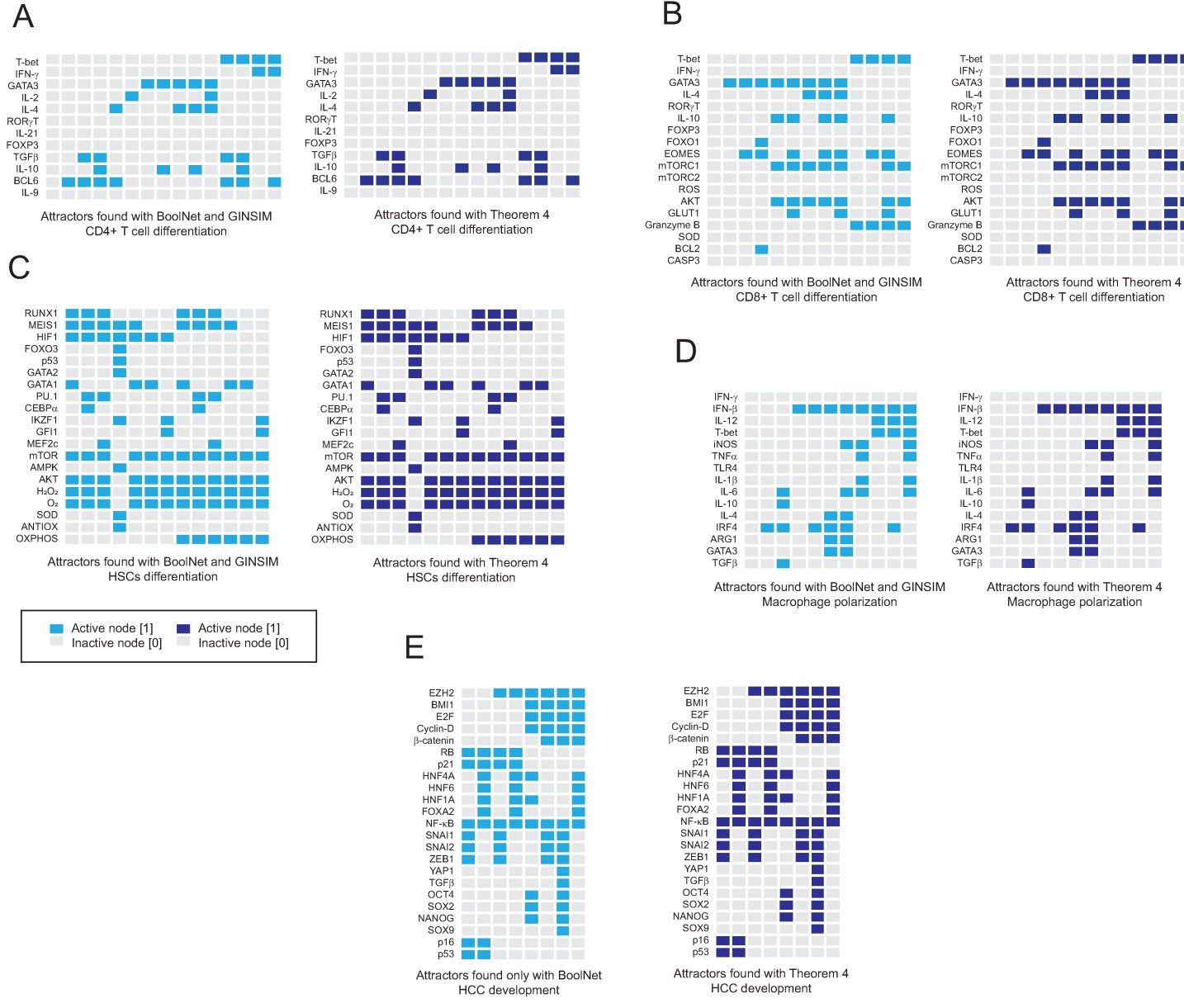

**Fig 9. Comparative results of the search for fixed-point attractors.** In this figure we show the result of using BoolNet [16] and GINSIM [17] *versus* the computational implementation in C# of Theorem 4 to search for fixed point attractors in real biological networks. Panel (A) presents the attractors obtained from a 12-node network representing CD4＋T cell differentiation, obtained from [8]. Panel (B) presents the attractors obtained from an 18-node network representing CD8＋T cell differentiation, obtained from [7]. Panel (C) presents the attractors of a 20-node network representing hematopoietic stem cell differentiation, obtained from [26]. Panel (D) shows the attractors of a 15-node network representing macrophage polarization, obtained from [27]. Finally, panel (E) shows the attractors of a 23-node network that models the development of hepatocellular carcinoma through epithelial-to-mesenchymal transition [9]. In all cases, the attractors obtained were identical, which validates our theoretical approach. Note: The only network for which GINSIM could not be used was the epithelial-mesenchymal transition network due to the size of the code used in this network.

Next, by using Theorem 4 on each module, we found the semi-attractors $\{0, 0, 0\}$ and $\{1, 1, 1\}$ for each module. Then, we used the procedure performed in Example 5 to calculate the canonical form of each module. As a result of this, we obtained:

$$
\begin{cases} x_1^+ = x_1 \\ x_2^+ = x_1 \\ x_3^+ = x_1 \end{cases}
\qquad
\begin{cases} x_4^+ = x_4 \\ x_5^+ = x_4 \\ x_6^+ = x_4 \end{cases}
\qquad
\begin{cases} x_7^+ = x_4 \\ x_8^+ = x_4 \\ x_9^+ = x_4 \end{cases}
$$

Note that for module $\{x_1,\ x_2,\ x_3\}$ there are only $2^3 = 8$ initial configurations, and according to their canonical form, the attractors $\{000\}$ and $\{111\}$ depend exclusively on the value of $x_1$. Therefore, the 4 configurations with $x_1 = 0$ will go to the attractor $\{000\}$ and the other 4 with $x_1 = 1$ will go to $\{111\}$. For modules $\{x_4,\ x_5,\ x_6\}$ and $\{x_7,\ x_8,\ x_9\}$, it is exactly the same situation, but now the dependence is on the value of $x_4$ instead of $x_1$. Joining the semi-attractors for each module, we obtain the configurations:

$$\{000000000\},\ \{000000111\},\ \{000111000\},\ \{000111111\},$$

$$\{111000000\},\ \{111000111\},\ \{111111000\},\ \{111111111\}.$$

Using the full network (E. 19) with Theorem 4 to determine which of such configurations are global attractors, we found the following true attractors:

$$\{000000000\},\ \{000111111\},\ \{111000000\},\ \{111111111\}$$

Regarding the basins of attraction of each attractor, the canonical form of each module indicates that all configurations with $x_1 = 0$ and $x_4 = 0$ will reach the attractor $\{000000000\}$, and there will be a total of $4 \times 4 \times 4 = 128$ configurations. Likewise, all configurations with $x_1 = 0$ and $x_4 = 1$, which will be 128, will converge towards the attractor $\{000111111\}$. While 128 configurations with $x_1 = 1$ and $x_4 = 0$ will reach the attractor $\{111000000\}$. Similarly, the 128 configurations with $x_1 = 1$ and $x_4 = 1$ will necessarily reach the attractor $\{111111111\}$. These results were corroborated using Visual Studio Community 2022 (see Supplementary Material).

Finally, we apply this methodology to study the dynamics of a well-known 200-node network. Since it is difficult to calculate the basins of attraction of a 200-node network, given that its state space has $2^{200} = 1.60694 \times 10^{60}$ initial configurations, we decided to expand the network (E. 19) to 200 nodes, given that this network has known dynamics. Specifically, we transformed the network (E. 19) as follows:

$$
\begin{cases}
x_1^+ = x_1 \\
x_2^+ = x_1 \\
x_3^+ = x_2 \\
x_4^+ = x_4 \\
x_5^+ = x_4 \\
x_6^+ = x_5 \\
x_7^+ = x_6 \\
x_8^+ = x_7 \\
\quad \vdots \\
x_{199}^+ = x_{198} \\
x_{200}^+ = x_{199}
\end{cases}
\tag{E. 20}
$$

We apply the methodology described in this section, but now we delimit 10 modules of 20 nodes each. After finding the semi-attractors for all modules, we proceeded to calculate the canonical form of each module. Subsequently, we joined the semi-attractors and found 2048 candidates for global attractors. As in the case of the network (E. 19), for

the module $\{x_1, x_2, x_3, x_4, \ldots, x_{20}\}$ the value of $x_1$ and $x_4$ determined that the semi-attractors $\{00000000000000000000\}$, $\{00011111111111111111\}$, $\{11100000000000000000\}$, $\{11111111111111111111\}$ were reached. For the rest of the modules, only the attractors $\{00000000000000000000\}$ and $\{11111111111111111111\}$ were found, which depended entirely on the value of $x_4$. Finally, we used the network (E. 20) with Theorem 4 to identify the true attractors, which were equivalent to the 4 attractors of the network (E. 19). In this case, the size of the basins of attraction of each of these attractors was $2^{58} = 2.8823 \times 10^{17}$ configurations, and the reachability conditions also depended on the values of $x_1$ and $x_4$. This experiment with a toy network demonstrates that the theory developed in this work can be used to calculate the reachability conditions of large-scale networks.

## Discussion

For many years, large Boolean networks have been reduced to smaller versions by finding the minimal, necessary, and sufficient components to generate their entire dynamics [4,8,30]. This is achieved by collapsing linear interactions while keeping nonlinear motifs present in their structures [31–33]. Although, this approach is useful in practice, it is not free from criticism, because some argue that it is difficult to prove that reduction does not lose information of the network [33]. However, the concept of the "*canonical form*" of Boolean networks is precisely the set of minimal, necessary, and sufficient elements to generate the entire dynamics of any network [14]. Originally, it was proposed that the canonical form of Boolean networks can be obtained by computing the product of matrix semi-tensors [14]. Nevertheless, Wang and colleagues proposed a simpler method to obtain the same information provided by the canonical form [15]. In this method, they used the logic rules of the network to determine the immediate output of each element of the state space, and subsequently the configurations of the states are converted into integers in an arrangement defined as a "*truth matrix*" [15].

Herein, we observed that the approach proposed by Wang and colleagues works to compute all the dynamical features of a Boolean network, because the successive composition of such a map is the synchronous update scheme (Figs 3 and 6). That is, all the features of a Boolean network are summarized in that initial map. By formalizing this idea, we observed that the equation of the temporal evolution of the synchronous update scheme (E. 3), corresponds to the matrix form of the Chapman-Kolmogorov equation [22], which has several important implications. First, the synchronous update scheme itself is equivalent to a particular case of Markovian chains, since there is only one probability of jumping to another state, which recontextualizes the nature of fixed-point attractors and cyclic attractors. The second direct implication is that, other approaches such as variational Bayesian inference and the master equation can be applied to formally study Boolean networks that are updated synchronously. In particular, variational Bayesian inference can be applied to generate analytical approximations to systems in which there are unobservable variables [34], which might allow working with a certain degree of precision with massive sequencing data, as well as with in *vivo* and real-time biological systems.

Notably, the results presented in this paper demonstrate that our approach of plotting the landscape of attractors in the Cartesian plane [13] is by itself the graphical representation of the canonical form of a Boolean network that only have fixed-point attractors (Fig 1). Herein we also proposed two methods to calculate the canonical form of any synchronously updated Boolean network. The first one consists in evaluating the expression (E. 3) as $t \to \infty$, verifying that it preserves its cyclic attractors as the function at time $t = 0$, and applying any formalism to know the canonical form of a logical function, such as calculating the disjunctive normal form (Fig 6). The second method consists in modifying the map of the Boolean network, making all the elements of the basins of attraction converge directly to the attractors, without altering the interactions that generate them (Fig 7).

After studying the synchronous update scheme, we tried to apply our analytical approach with the asynchronous update scheme. We were motivated to do this, because it has been postulated that the asynchronous update scheme is more realistic than the synchronous update strategy, under the assumption that genes and biological processes do not occur at the same time [11]. Despite its wide acceptance in the biological community, this type of update scheme has *sui generis* and unexplained dynamical features. For instance, it is difficult for a cyclic attractor to be preserved [25], there are

no perfectly defined basins of attraction, since they might change in size and in element composition [25]. In this context, fixed-point attractors are always maintained regardless of the chosen update order, so they are believed to be robust and biologically meaningful [11]. Beyond all these curious observations, there is no logical explanation of what causes them. Herein, we found that the update order of the asynchronous scheme generates a series of functions different from the original structure of the network (Fig 8). We also found that, the periodic composition of such functions is the asynchronous update scheme *per se*. Interestingly, we noted that these alternative functions harm the interactions that generate cyclic attractors, except for fixed-point attractors. We also observed that the order in which the composition of the functions is performed might affect the size and composition of the elements of the basins of attraction (Fig 8). Together, these results finally clarify the origin of all the unexplained features of the asynchronous update scheme.

Furthermore, we showed that it is not possible to create a canonical form for networks that are updated asynchronously (Fig 8), and its study is only applicable to the synchronous update scheme. At the moment, we do not know whether it is adequate to assume that the expression time of a gene is equivalent to the order and sequence of updates of a node, but our results suggest that it is important to reexamine the philosophical foundations of such an assumption and to look for a new way to physically interpret the asynchronous update scheme.

## Conclusions

In conclusion, in this work we demonstrated that the synchronous update scheme is a particular case of Markov chains. We also found that the landscape of attractors plotted in the Cartesian plane is a graphical representation of the solution of matrix equation of Chapman-Kolmogorov, and it is the canonical form of certain types of Boolean networks. Also, we showed that it is not possible to apply these formalisms to the asynchronous update scheme. Furthermore, we found that the root of all the *sui generis* dynamical features of the asynchronous update scheme is due to the composition of functions that originates it, so it is recommended to reconsider its use in the study of biological systems.

### Glossary

**Attractor Landscape**: This concept refers to a set of attractors proper of a dynamical system. In this paper, we used this term to refer to the set of fixed-point and cyclic attractors of a Boolean network.

**Basin of Attraction**: This concept refers to the set of all initial states that converges toward a specific attractor in a Boolean network.

**Canonical Form**: In mathematics this concept refers to a standardized or normalized representation of an equation, which allows for easier comparison and analysis.

**Cyclic attractor**: This concept refers to a set of states in a dynamic system through which the system cycles indefinitely in a period.

**Disjunctive Normal Form (DNF)**: This is a standard way of writing a Boolean expression as a sum of min-terms.

**Fixed-point attractor**: This concept refers to a stable state of a dynamic system where the system remains once it is reached. Formally: $f(p) = p$, $\forall t$, where $p$ in a fixed-point attractor.

**Function**: This is a special type of relation that assigns exactly one output to each input from a given domain ($D$). Formally: $\forall x \in D$, $\exists! y \in C : f(x) = y$, where $C$ is the codomain.

**Function Composition**: This concept refers to the process of applying one function to the result of another. Formally: $(f \circ g)(x) = f(g(x))$, where $f$ and $g$ are functions.

**Min-terms**: This concept refers to the simplest form of expression in Boolean algebra, which represents a unique combination of logic variables that yields a value of true, or numerically 1.

**Reflexivity**: This concept refers to a property of a binary relation in which every element is related to itself. Formally: $\forall a$, $(a, a) \in R$.

**Relation**: This term refers to a logical or mathematical connection between elements of two or more sets, often used to define functions, mappings, or other associations.

## Supporting information

**S1 Text. Software appendix.** This file includes all the codes for the networks used in this work, including their implementation in C#, BoolNet, and SMBL for GINSIM.
(DOCX)

## Acknowledgments

AB thanks to SECIHTI and Cinvestav-IPN for his designation as "Investigador por México". JAAG thanks to SECIHTI for his doctoral grant.

## Author contributions

**Conceptualization:** Antonio Bensussen, Juan Carlos Martínez-García.

**Formal analysis:** Antonio Bensussen.

**Resources:** Juan Carlos Martínez-García.

**Software:** J. Arturo Arciniega-González.

**Supervision:** Juan Carlos Martínez-García.

**Validation:** J. Arturo Arciniega-González.

**Visualization:** Antonio Bensussen.

**Writing – original draft:** Antonio Bensussen.

**Writing – review & editing:** Antonio Bensussen, Elena R. Álvarez-Buylla, Juan Carlos Martínez-García.

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
