## [Decision Letter · Decision Letter 0]

12 Mar 2025

PONE-D-25-05122Analytical approach of synchronous and asynchronous update schemes applied to solving biological Boolean networksPLOS ONE

Dear Dr. Martinez-Garcia,

Thank you for submitting your manuscript to PLOS ONE. After careful consideration, we feel that it has merit but does not fully meet PLOS ONE’s publication criteria as it currently stands. Therefore, we invite you to submit a revised version of the manuscript that addresses the points raised during the review process. Both referees found some major issues in the consistency of the results and their interpretation. A truly major revision is needed to address all of the raised major issues. Please submit your revised manuscript by Apr 26 2025 11:59PM. If you will need more time than this to complete your revisions, please reply to this message or contact the journal office at plosone@plos.org . Please include the following items when submitting your revised manuscript:

We look forward to receiving your revised manuscript.

Kind regards,

Attila Csikász-Nagy

Academic Editor

PLOS ONE

Journal Requirements:

“Elena R. Álvarez-Buylla and Juan Carlos Martínez-García acknowledge the support from UNAM-DGAPA PAPIIT IN211721 “Patrones genéricos y sistémicos de la diferenciación y la proliferación en los nichos de células troncales: Raíz de Arabidopsis thaliana como sistema de estudio teórico-experimental” and CONACYT-FORDECYT-PRONACES 194186/2020 “Biología matemática y computacional de sistemas médicos: modulación preventiva de la emergencia y progresión de enfermedades crónico-degenerativas.”, respectively. “

“AB thanks to CONAHCYT and Cinvestav-IPN for his designation as “Investigador por México”. JAAG thanks to CONAHCYT for his doctoral grant. Elena R. Álvarez-Buylla and Juan Carlos Martínez-García acknowledge the support from UNAM-DGAPA PAPIIT 23 IN211721 “Patrones genéricos y sistémicos de la diferenciación y la proliferación en los nichos de células troncales: Raíz de Arabidopsis thaliana como sistema de estudio teórico[1]experimental” and CONACYT-FORDECYT-PRONACES 194186/2020 “Biología matemática y computacional de sistemas médicos: modulación preventiva de la emergencia y progresión de enfermedades crónico-degenerativas.”, respectively.”

“Elena R. Álvarez-Buylla and Juan Carlos Martínez-García acknowledge the support from UNAM-DGAPA PAPIIT IN211721 “Patrones genéricos y sistémicos de la diferenciación y la proliferación en los nichos de células troncales: Raíz de Arabidopsis thaliana como sistema de estudio teórico-experimental” and CONACYT-FORDECYT-PRONACES 194186/2020 “Biología matemática y computacional de sistemas médicos: modulación preventiva de la emergencia y progresión de enfermedades crónico-degenerativas.”, respectively. “

4. Please amend the manuscript submission data (via Edit Submission) to include author Dr. Elena R. Álvarez-Buylla

5. Please amend your authorship list in your manuscript file to include author Dr. María Álvarez-Buylla Roces.

Reviewers' comments:

Reviewer's Responses to Questions

**Comments to the Author**

1. Is the manuscript technically sound, and do the data support the conclusions?

Reviewer #1: No

Reviewer #2: Yes

2. Has the statistical analysis been performed appropriately and rigorously? 

Reviewer #1: N/A

Reviewer #2: N/A

3. Have the authors made all data underlying the findings in their manuscript fully available?

Reviewer #1: No

Reviewer #2: Yes

4. Is the manuscript presented in an intelligible fashion and written in standard English?

Reviewer #1: No

Reviewer #2: Yes

5. Review Comments to the Author

Reviewer #1: This manuscript builds on prior work by Wang et al and by the authors to obtain the “canonical form” of a Boolean network by translating the landscape of attractors into the Cartesian plane. Unfortunately, the manuscript is not clear and understandable to the average reader. (This reviewer spent several hours trying to understand the manuscript, with limited success.) Partly due to the lack of clarity, many of the claims made in the abstract are unsupported. Most importantly, the single paragraph dedicated to solving real networks of biological systems does not describe which of the methods were actually applied, and how.

Major concerns

1. The manuscript needs a section with definitions, explanation of notations, and background information. For example, the representation of Boolean functions used in the manuscript needs to be explained. Technical terms such as “relation”, “reflexive element” need to be defined. In the statement of Theorem 3 it is not clear what is meant by c1Rc2 ^ c2Rc3 ^c3Rc1. This makes the theorem unintelligible. Moreover, the authors should not assume that every reader knows what f o f means.

2. The point that the synchronous update scheme of a Boolean network is a particular case of Markov chains has been known for a while. For example, Ilya Shmulevich’s book on Probabilistic Boolean Networks uses Markov chains extensively.

3. The application of the theories is very limited. Instead of implementing the involved methodologies of obtaining the canonical form of the Boolean system, from the single paragraph dedicated to the application it seems that the only thing done was to determine the fixed point attractors of the systems. Determining fixed point attractors can be done by multiple means. Thus, the practical utility of the work presented in the manuscript is not clear.

4. The Supporting Information was not available for review.

5. The paragraph of the Discussion dedicated to asynchronous update needs extensive revision. It is not factually correct that the dynamic features of asynchronous update are unexplained. The study of asynchronous Boolean systems is at least 60 years old, René Thomas being a very active pioneer. The degree of preservation of cyclic attractors of synchronous update has been well studied in the literature. Asynchronous update in which only one node may change at a time can also lead to a cyclic attractor. A necessary condition for such an attractor is the existence of a negative feedback loop in the interaction network (as conjectured by Rene Thomas and proven later). Cyclic attractor that rely on a positive feedback loop and node synchrony are not preserved. Various flavors of asynchronous and noisy update are actively researched, to establish the time implementation most suited for biological systems. There is a significant effort to find the update-independent features of Boolean systems, for example, their trap spaces.

More minor and specific points

6. The goal of finding the minimal necessary and sufficient components to generate a system’s dynamics is stated as a motivation for determining the canonical form. Yet, the manuscript does not establish how the canonical form may lead to the information on the necessary/sufficient components.

7. “Topology of a network” is often used to refer to the graph structure of the network (the distribution of nodes and edges). “Topology of a Boolean network” can be misinterpreted to refer to the structure of the interaction network that underlies the Boolean system. The authors seem to mean the dynamics of the Boolean network, or its attractor landscape. They should state it accordingly.

8. The intuition behind Lemma 1 and the function H needs to be presented in words. In particular, the reader may miss that u refers to the basin of attraction of attractor p. Example 1 needs more explanation, especially figure 1B, which illustrates the foundational concept, the function that corresponds to the landscape of attractors. The landscape of attractors is defined in line 73 as the set of attractors. It is not clear that the function also refers to the attractor basins. No description is given on how the disjunctive normal form is obtained, what its purpose is.

9. In line 199, what does S(u0, t=0) mean? Why does it equal u0 S(u0, t=1)? None of the equalities stated on that line make sense without definitions and explanations.

Reviewer #2: The paper addresses an important issue in modeling, especially biochemical systems - the reachability analysis of possible system states, which is still a challenging problem. The authors consider asynchronous and synchronous simulation for Boolean networks. The paper presents synchronous update schemes to characterize the topology of Boolean networks. The authors describe several theorems, which they prove and illustrate using small examples. They show a new matrix-based method to solve synchronously updated Boolean networks and derive interesting properties for Boolean networks. The authors show that the landscape of attractors of any Boolean network can be visualized using integers in the Cartesian plane, which allows one to easily obtain the canonical form of Boolean networks. They found an unexpected connection between the landscape of attractors and Markov chains. Moreover, the authors provide a simplified method for computing the canonical form of synchronously updated networks. They consider a matrix formalism of an asynchronous update scheme.

The paper is well-written and clearly explained.

I find the approach interesting, but I see some weaknesses, so I recommend a major revision.

(1) Please provide the biological models in the SBML format for reuse by the community and by other software tools

(2) Refer to related work, such as other tools, e.g., GINSIM.

(3) How does the approach relate to minimal transition invariants in Petri nets? Minimal transition invariants in Petri nets characterize the minimum number of nodes necessary for the functioning of the network, which seems to be similar to the concept of the canonical form of Boolean networks that should be precisely the set of minimal, necessary, and sufficient elements to generate the entire dynamics of any network.

(4) The biological examples are rather small networks. Say something about the scalability of your approach.

6. PLOS authors have the option to publish the peer review history of their article (what does this mean? ). If published, this will include your full peer review and any attached files.

**Do you want your identity to be public for this peer review?** For information about this choice, including consent withdrawal, please see our Privacy Policy .

Reviewer #1: No

Reviewer #2: No

---

## [Author Response · Author response to Decision Letter 1]

17 Jun 2025

Journal Requirements

Requirements 1, 2, 3, 4 5, and 6, have been taken into account.

Review Comments to the Author

Because we include notation that cannot be displayed here, please refer to the corresponding file we are submitting with this new version of our paper.

---

## [Decision Letter · Decision Letter 1]

12 Aug 2025

PONE-D-25-05122R1Analytical approach of synchronous and asynchronous update schemes applied to solving biological Boolean networksPLOS ONE

Dear Dr. Martinez-Garcia,

Thank you for submitting your manuscript to PLOS ONE. After careful consideration, we feel that it has merit but does not fully meet PLOS ONE’s publication criteria as it currently stands. Therefore, we invite you to submit a revised version of the manuscript that addresses the points raised during the review process. Reviewer 1 has some suggestions for rewording and explanation of details. Please update the text with taken these in consideration.

We look forward to receiving your revised manuscript.

Kind regards,

Attila Csikász-Nagy

Academic Editor

PLOS ONE

Journal Requirements:

Reviewers' comments:

Reviewer's Responses to Questions

**Comments to the Author**

1. If the authors have adequately addressed your comments raised in a previous round of review and you feel that this manuscript is now acceptable for publication, you may indicate that here to bypass the “Comments to the Author” section, enter your conflict of interest statement in the “Confidential to Editor” section, and submit your "Accept" recommendation.

Reviewer #1: All comments have been addressed

Reviewer #2: All comments have been addressed

2. Is the manuscript technically sound, and do the data support the conclusions?

Reviewer #1: Yes

Reviewer #2: Yes

3. Has the statistical analysis been performed appropriately and rigorously? 

Reviewer #1: N/A

Reviewer #2: N/A

4. Have the authors made all data underlying the findings in their manuscript fully available?

Reviewer #1: Yes

Reviewer #2: Yes

5. Is the manuscript presented in an intelligible fashion and written in standard English?

Reviewer #1: Yes

Reviewer #2: Yes

6. Review Comments to the Author

Reviewer #1: The revised manuscript addresses my points.

There is a single remaining sentence, on line 385-386, that is unclear

"Furthermore, it was surprising that only fixed-point attractors were found in the synchronous and

asynchronous update schemes (27)."

What subset of Boolean models is referred to here? It certainly cannot be all Boolean dynamical systems. Why was this surprising? Please rephrase to the sentence is clear and to the point.

Reviewer #2: The authors addressed all my comments in a correct way. I have no comments and suggest the publication of the paper.

7. PLOS authors have the option to publish the peer review history of their article (what does this mean? ). If published, this will include your full peer review and any attached files.

**Do you want your identity to be public for this peer review?** For information about this choice, including consent withdrawal, please see our Privacy Policy .

Reviewer #1: No

Reviewer #2: No

---

## [Author Response · Author response to Decision Letter 2]

20 Aug 2025

Reply to reviewer 1:

Reviewer #1: The revised manuscript addresses my points.

There is a single remaining sentence, on line 385-386, that is unclear

"Furthermore, it was surprising that only fixed-point attractors were found in the synchronous and asynchronous update schemes (27)."

What subset of Boolean models is referred to here? It certainly cannot be all Boolean dynamical systems. Why was this surprising? Please rephrase to the sentence is clear and to the point.

Dear reviewer,

in attention to your request, we rewrote the sentence to clarify that we were referring to Boolean networks.

Specifically, we replaced "Furthermore, it was surprising that only fixed-point attractors were found in the synchronous and asynchronous update schemes" with "These studies also showed that fixed-point attractors are invariant to the update scheme used to analyze Boolean networks".

These changes were implemented on lines 373-374.

---

## [Editor Report · Decision Letter 2]

22 Aug 2025

Analytical approach of synchronous and asynchronous update schemes applied to solving biological Boolean networks

PONE-D-25-05122R2

Dear Dr. Martinez-Garcia,

We’re pleased to inform you that your manuscript has been judged scientifically suitable for publication and will be formally accepted for publication once it meets all outstanding technical requirements.

Kind regards,

Attila Csikász-Nagy

Academic Editor

PLOS ONE
---

## [Editor Report · Acceptance letter]

PONE-D-25-05122R2

PLOS ONE

Dear Dr. Martinez-Garcia,

I'm pleased to inform you that your manuscript has been deemed suitable for publication in PLOS ONE. Congratulations! Your manuscript is now being handed over to our production team.

Kind regards,

on behalf of

Dr. Attila Csikász-Nagy

Academic Editor

PLOS ONE